# GPS-SSL: Guided Positive Sampling to Inject Prior Into Self-Supervised Learning

## Abstract

In this paper, we propose *Guided Positive Sampling Self-Supervised Learning* (GPS-SSL), a general method to embed a priori knowledge into Self-Supervised Learning (SSL) positive samples selection. Current SSL methods leverage Data-Augmentations (DA) for generating positive samples and their performance heavily relies on the chosen set of DA. However, designing optimal DAs given a target dataset requires domain knowledge regarding that dataset and can be costly to search and find. Our method designs a metric space where distances better align with semantic relationship thus enabling nearest neighbor sampling to provide meaningful positive samples. This strategy comes in contrast with the current strategy where DAs are the sole mean to incorporate known properties into the learned SSL representation. A key benefit of GPS-SSL lies in its applicability to any SSL method, e.g. SimCLR or BYOL. As a direct by-product, GPS-SSL also reduces the importance of DA to learn informative representations, a dependency that has been one of the major bottlenecks of SSL. We evaluate GPS-SSL along with multiple baseline SSL methods on multiple downstream datasets from different domains when the models use *strong* or *minimal* data augmentations. We show that when using strong DAs, GPS-SSL outperforms the baselines on understudied domains. Additionally, when using minimal augmentations –which is the most realistic scenario for which one does not know a priori the strong DA that aligns with the possible downstream tasks– GPS-SSL outperforms the baselines on all datasets by a significant margin. We believe that opening a new avenue to impact the SSL representations that is not solely based on altering the DA will open the door to multiple interesting research directions, greatly increasing the reach of SSL.

## 1 Introduction

Self-supervised learning (SSL) has recently shown to be one of the most effective learning paradigms across many data domains (Radford et al., 2021; Girdhar et al., 2023; Assran et al., 2023; Chen et al., 2020; Grill et al., 2020; Bardes et al., 2021; Balestriero et al., 2023). SSL belongs to the broad category of annotation-free representation learning approaches, which have enabled machine learning models to use abundant and easy-to-collect unlabeled data, facilitating the training of ever-growing deep neural network architectures.

Despite the SSL promise, current approaches require handcrafted a priori knowledge to learn useful representations. This a priori knowledge is often injected through the positive sample – *i.e.*, semantically related samples – generation strategies employed by SSL methods (Chen et al., 2020). In fact, SSL representations are learned so that such positive samples get as similar as possible in embedding space, all while preventing a collapse of the representation to simply predicting a constant for all inputs. The different strategies to achieve that goal lead to different flavors of SSL methods (Chen et al., 2020; Grill et al., 2020; Bardes et al., 2021; Zbontar et al., 2021; Chen & He, 2021). In computer vision, positive sample generation mostly involves sampling an image from the dataset, and applying multiple handcrafted and heavily tuned data augmentations (DAs) to it, such as rotations and random crops, which preserve the main content of the image.

The importance of selecting the right DAs is enormous as it impacts performances to the point of producing a near random representation, in the worst case scenario (Balestriero et al., 2023).

As such, tremendous time and resources have been devoted to designing optimal DA recipes, most notably for eponymous datasets such as ImageNet (Deng et al., 2009). From a practitioner's standpoint, positive sample generation could thus be considered solved if one were to deploy SSL methods *only* on such popular datasets. Unfortunately – and as we will thoroughly demonstrate throughout this paper –, common DA recipes used in those settings fail to transfer to other datasets. We hypothesize that as the dataset domains get semantically further from ImageNet, on which the current set of optimal DAs are designed, the effectiveness of DAs reduces. For example, since ImageNet consists of natural images, mostly focused around 1000 different object categories, we observe and report this reduction of performance on datasets consisting of more specialized images, such as hotel room images (Stylianou et al., 2019; Kamath et al., 2021), or only focus on different types of airplanes (Maji et al., 2013), or medical images (Yang et al., 2023). Since searching for the optimal DAs is computationally intense, there remains an important bottleneck when it comes to deploying SSL to new or under-studied datasets. This becomes in particular important when applying SSL methods on data gathered for real-world applications.

In this paper, we introduce a strategy to obtain positive samples, which generalizes the well established NNCLR SSL method (Dwibedi et al., 2021). While NNCLR proposes to obtain positive samples by leveraging known DAs and nearest neighbors in the embedding space of the network being trained, we propose to perform nearest neighbour search in the embedding space of a pre-defined mapping of each image to its possible positive samples. The mapping may generated by a clone of the network being trained – therefore recovering NNCLR – but perhaps most interestingly may also be generated by any pre-trained network or even hand-crafted. This flexibility allows to (i) enable simple injection of prior knowledge into positive sampling –without relying on tuning the DA– and most importantly (ii) makes the underlying SSL method much more robust to under-tuned DAs parameters. By construction, the proposed method – coined GPS-SSL for Guided Positive Sampling Self-Supervised Learning–, can be coupled off-the-shelf with any SSL method used to learn representations, *e.g.*, BarlowTwins (Zbontar et al., 2021), SimCLR (Chen et al., 2020), BYOL (Grill et al., 2020). We validate the proposed GPS-SSL approach on a benchmark suite of under-studied datasets, namely FGVCAircraft, PathMNIST, TissueMNIST, and show remarkable improvements over baseline SSL methods. We further evaluate our model on a real-world dataset, Revised-Hotel-ID (R-HID) (Feizi et al., 2022) and show clear improvements of our method compared the baseline SSL methods. Finally, we validate the approach on commonly used image datasets with known effective DAs recipes, and show that GPS remains competitive. Through comprehensive ablations, we show that GPS-SSL takes a step towards shifting the focus of designing *well-crafted DAs* to having a better *prior knowledge* embedding space in which choosing the nearest neighbour becomes an attractive positive sampling strategy.

The main contributions of this paper can be summarized as follows:

- We propose a positive sampling strategy, Guided Positive Sampling Self-Supervised Learning (GPS-SSL) that enables SSL models to use prior knowledge about the target-dataset to help with the learning process and reduce the reliance on carefully hand-crafted augmentations. The prior knowledge is a mapping between images and a few of their closest nearest neighbors that could be calculated with a pre-trained network or even hand-crafted.

- Moreover, we evaluate GPS-SSL by applying it to baseline SSL methods and show that with strong augmentations, they perform comparable to, or better than, the original methods. Moreover, they significantly outperform the original methods when using minimal augmentations, making it suitable for learning under-studied or real-world datasets (rather than transfer-learning).

- To further evaluate our model on datasets with under-studied applications, we consider hotel retrieval task in the counter human trafficking domain. Similar to benchmark datasets, we see on this less studied dataset, our proposed GPS-SSL outperforms the baseline SSL methods by a significant margin.

We provide the code for GPS-SSL and downloading and using R-HID on GitHub, available at: https://anonymous.4open.science/r/gps-ssl-1E68, for the research community.

## 2   RELATED WORK

Self Supervised Learning (SSL) is a particular form of unsupervised learning methods in which a given Deep Neural Network (DNN) learns meaningful representations of their inputs without labels.

The variants of SSL are numerous. At the broader scale, SSL defines a pretext task on the input data and train themselves by solving the defined task. In SSL for computer vision, the pretext tasks generally involve creating different views of images and encoding both so that their embeddings are close to each other. However, that criteria alone would not be sufficient to learning meaningful representations as a degenerate solution is for the DNN to simply collapse all samples to a single embedding vector. As such, one needs to introduce an "anti-collapse" term. Different types of solutions have been proposed for this issue, splitting SSL methods into multiple groups, three of which are: 1) Contrastive(Chen et al., 2020; Dwibedi et al., 2021; Kalantidis et al., 2020): this group of SSL methods prevent collapsing by considering all other images in a mini-batch as negative samples for the positive image pair and generally use the InfoNCE (Oord et al., 2018) loss function to push the negative embeddings away from the positive embeddings. 2) Distillation(Grill et al., 2020; He et al., 2020; Chen & He, 2021): these methods often have an asymmetric pair of encoders, one for each positive view, where one encoder (teacher) is the exponential moving average of the other encoder (student) and the loss only back-propagates through the student encoder. In general, this group prevents collapsing by creating asymmetry in the encoders and defines the pre-text task that the student encoder must predict the teach encoder's output embedding. 3) Feature Decorrelation(Bardes et al., 2021; Zbontar et al., 2021): These methods focus on the statistics of the embedding features generated by the encoders and defines a loss function to encourage the embeddings to have certain statistical features. By doing so, they explicitly force the generated embeddings not to collapse. For example, Bardes et al. (2021) encourages the features in the embeddings to have high variance, while being invariant to the augmentations and also having a low covariance among different features in the embeddings. Besides these groups, there are multiple other techniques for preventing collapsing, such as clustering methods (Caron et al., 2020; Xie et al., 2016), gradient analysis methods (Tao et al., 2022).

Although the techniques used for preventing collapse may differ among these groups of methods, they generally require the data augmentations to be chosen and tuned carefully in order to achieve high predictive performance (Chen et al., 2020). Although choosing the optimal data augmentations and hyper-parameters may be considered a solved problem for popular datasets such as Cifar10 (Krizhevsky et al., 2009) or ImageNet (Deng et al., 2009), the SSL dependency on DA remains their main limitation to be applied to large real-world datasets that are not akin natural images. Due to the importance of DA upon the DNN's representation quality, a few studies have attempted mitigation strategies. For example, Cabannes et al. (2023b) ties the impact of DA with the implicit prior of the DNN's architecture, suggesting that informed architecture may reduce the need for well designed DA although no practical answer was provided. Cabannes et al. (2023a) proposed to remove the need for DA at the cost of requiring an oracle to sample the positive samples from the original training set. Although not practical, this study brings a path to train SSL without DA. Additionally, a key limitation with DA lies in the need to be implemented and fast to produce. In fact, the strong DA strategies required by SSL are one of the main computational time bottleneck of current training pipelines (Bordes et al., 2023). Lastly, the over-reliance on DA may have serious fairness implications since, albeit in a supervised setting, DA was shown to impact the DNN's learned representation in favor of specific classes in the dataset (Balestriero et al., 2022).

All in all, SSL would greatly benefit from a principled strategy to embed a priori knowledge into generating positive pairs that does not rely on DA. We propose a first step towards such Guided Positive Sampling (GPS) below.

## 3   GUIDED POSITIVE SAMPLING FOR SELF-SUPERVISED LEARNING

We propose a novel strategy, *Guided Positive Sampling Self-Supervised Learning* (GPS-SSL), that takes advantage of prior knowledge for positive sampling to make up for the sub-optimality of generating positive pairs solely from DA in SSL.

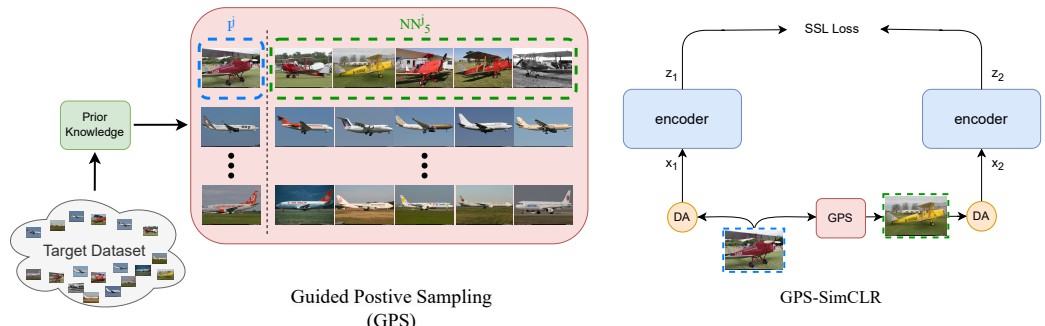

Figure 1: Our strategy, GPS-SSL, for positive sampling based on prior knowledge DA-based methods.

### 3.1 GPS-SSL: NEAREST NEIGHBOR POSITIVE SAMPLING IN ANY DESIRED EMBEDDED SPACE

As theoretically shown in various studies (HaoChen et al., 2021; Balestriero & LeCun, 2022; Kiani et al., 2022), the principal factors that impacts the quality of the learned representation resides in how the positive pairs are defined. In fact, we recall that in all generality, SSL losses that are minimized can mostly be expressed as

$$\mathcal{L}_{\text{SSL}} = \sum_{(\boldsymbol{x},\boldsymbol{x}') \in \text{PositivePairs}} \text{Distance}(f_\theta(\boldsymbol{x}), f_\theta(\boldsymbol{x}')) - \text{Diversity}(\{f_\theta(\boldsymbol{x}), \boldsymbol{x} \in \mathbb{X}\}), \quad (1)$$

for the current training or mini-batch $\mathbb{X}$, a distance measure such as the $\ell_2$ norm or the cosine similarity, and a diversity measure such that the rank of the embedings or proxies of their entropy. All in all, defining the right set of PositivePairs is what determines the ability of the final representation to solve downstream tasks. The common solution is repeatedly apply a DA onto a single datum to generate such positive pairs:

$$\text{PositivePairs} \triangleq \{(\text{DA}(\boldsymbol{x}), \text{DA}(\boldsymbol{x})), \forall \boldsymbol{x} \in \mathbb{X}\}, \quad (2)$$

where the DA operator includes the random realisation of the DA such as the amount of rotation or zoom being applied onto its input image. However that strategy often reaches its limits since such DAs need to be easily implemented for the specific data that is being used, and it needs to be known a priori. When considering an image dataset, the challenge of designing DA for less common datasets, e.g., FGVCAircraft, led practitioners to instead train the model on a dataset such as ImageNet, where strong DAs have already been discovered, and then transferring the model to other datasets. This however has its limits, e.g. when considering medical images.

As an alternative, we propose an off-the-shelf strategy to sample positive pairs that can be equipped onto any baseline SSL method, e.g., SimCLR, VICReg, coined GPS-SSL and which is defined by defining positive pairs through nearest neighbour sampling in an a priori known embedding space:

$$\text{PositivePairs}_{\text{GPS}} \triangleq \{(\text{DA}(\boldsymbol{x}), \text{DA}(\boldsymbol{x}')), \forall (\boldsymbol{x}, \boldsymbol{x}') \in \mathbb{X}^2 : \arg\max_{\boldsymbol{u} \in \mathbb{X}} \|g_\gamma(\boldsymbol{u}) - g_\gamma(\boldsymbol{x})\|_2^2 < \tau\}, \quad (3)$$

for some positive value $\tau$. In short, replace the set of positive pairs generated from applying a given DA to a same input, by applying a given DA onto two different inputs found so that one is the nearest neighbor of the other in some embedding space provided by $g_\gamma$. From this, we obtain a first direct result below making GPS-SSL recover a powerful existing method known as NNCLR.

**Proposition 1** *For any employed DA, GPS-SSL which replaces eq. (2) by eq. (3) in any SSL loss (eq. (1)) recovers (i) input space nearest neighbor positive sampling when $g_\gamma$ is the identity and $\tau \gg 0$, (ii) standard SSL when $g_\gamma$ is the identity but $\tau \to 0$, and (iii) NNCLR when $g_\gamma = f_\theta$ and $\tau \to 0$.*

The above result provides a first strong argument demonstrating how GPS-SSL does not reduce the capacity of SSL, in fact, it introduces a novel axis of freedom–namely the design of $(g_\gamma, \tau)$–to extend current SSL beyond what is amenable solely by tweaking the original architecture $f_\theta$, or the

original DA. In particular, the core motivation of the presented method is that this novel ability to design $g_\gamma$ also reduces the burden to design DA. In fact, if we consider the case where the original DA is part of the original dataset

$$\forall \boldsymbol{x} \in \mathbb{X}, \exists \rho : DA(\boldsymbol{x}; \rho) \in \mathbb{X}, \tag{4}$$

i.e., for any sample in the training set $\mathbb{X}$, there exists at least one DA configuration ($\rho$) that produces another training set sample, GPS-SSL can recover standard SSL albeit without employing any DA.

**Theorem 1** *Performing standard SSL (employing eq. (2) into eq. (1)) with a given DA and a training set for which eq. (4) holds, is equivalent to performing GPS-SSL (employing eq. (2) into eq. (1)) without any DA and by setting $g_\gamma$ to be invariant to that DA, i.e. $g_\gamma(DA(\boldsymbol{x})) = g_\gamma(\boldsymbol{x})$.*

By construction from eq. (4) and assuming that one has the ability to design such an invariant $g_\gamma$, it is clear that the nearest neighbour within the training set for any $\boldsymbol{x} \in \mathbb{X}$ will be the corresponding samples $DA(\boldsymbol{x})$ therefore proving theorem 1. That result is quite impractical but nevertheless provides a great motivation to GPS-SSL. Having the ability to design $g_\gamma$ not only has the ability to encompass the burden of design a DA, but both can be used jointly allowing one embed as much a priori knowledge as possible through both venues simultaneously.

**The design of $g_\gamma$.** The proposed strategy (eq. (3)) is based on finding the nearest neighbors of different candidate inputs in a given embedding space. There are multiple ways for acquiring an informative embedding space, i.e., a prescribed mapping $g_\gamma$. Throughout our study, we will focus on the most direct solution of employing a previously pretrained mapping. The pre-training may or may not have occurred on the same dataset being considered for SSL. Naturally, the alignment between both datasets affects the quality and reliability of the embeddings. If one does not have access to such pretrained model, another solution is to first learn an abstracted representation, e.g., an auto-encoder or VAE (Kingma & Welling, 2013), and then use the encoder for $g_\gamma$. In that setting the motivation lies in the final SSL representation being superior to solve downstream tasks that the encoder ($g_\gamma$) alone. We provide some examples of the resulting positive pairs with our strategy in Figure 1. In this figure, we use a pretrained model to calculate the set of $k$ nearest neighbors for each image $\boldsymbol{x}$ in the target dataset. Then for each image $\boldsymbol{x}$, the model randomly chooses the positive image from the nearest neighbors in embedding space (recall eq. (3)). Finally, both the original image and the produced positive sample are augmented using the chosen DA and passed as a positive pair of images through the encoders. Note that as per proposition 1, GPS-SSL may choose the image itself as its own positive sample, but the probability of it happening reduces as $\tau$ increases. As we will demonstrate the later sections, the proposed positive sampling strategy often outperforms the baseline DA based positive pair sampling strategy on multiple datasets.

**Relation to NNCLR** The commonality of NNCLR and GPS-SSL has been brought forward in proposition 1. In short, they both choose the nearest neighbor of input images as the positive sample. However, the embedding space in which the nearest neighbor is chosen is different; in NNCLR, the model being trained creates the embedding space which is thus updated at every training step, i.e., $g_\gamma = f_\theta$. However, GPS-SSL generalizes that in the sense that the nearest neighbors can stem from any prescribed mapping, without the constraint that it is trained as part of the SSL training, or even that it takes the form of a DNN. The fact that NNCLR only considers the model being trained to obtain its positive samples also makes it heavily dependent on complex and strong augmentations to produce non degenerate results. On the other hand, our ability to prescribe other mappings for the nearest neighbor search makes GPS-SSL much less tied to the employed DA. We summarize those methods in Figure 3.

## 3.2 EMPIRICAL VALIDATION ON BENCHMARKED DATASETS

In our experiments, we train the baseline SSL methods and the proposed GPS-SSL with two general sets of augmentations, *StrongAug*, which are augmentations that have been finetuned on the target dataset (for Cifar10 (Krizhevsky et al., 2009)) or ImageNet in the case of under-studied datasets (for FGVCAircraft (Maji et al., 2013), PathMNIST (Yang et al., 2023), TissueMNIST (Yang et al., 2023), and R-HID), and *RHFlipAug*, representing the scenario where we do not know the correct augmentations and use minimal ones. The set of *StrongAug* consists of `random-resized-crop`, `random-horizontal-flip`, `color-jitter`, `gray-scale`, `gaussian-blur` while `solarization` while *RHFlipAug* only uses `random-horizontal-flip`.

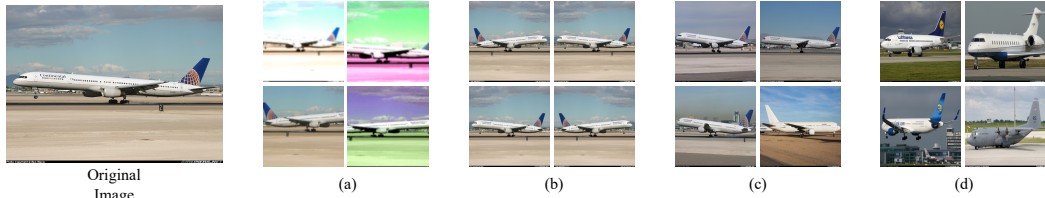

Figure 2: An example (a) *StrongAug* and (b) *RHFlipAug* applied to an image from the FGVCAircraft dataset. Furthermore, (c) and (d) depict examples of the 4 nearest neighors calculated by CLIP and VAE embeddings, respectively.

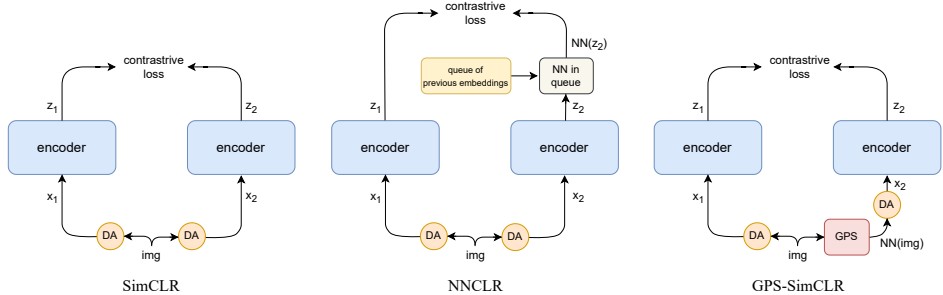

Figure 3: Architectures of SimCLR, NNCLR, and GPS-SimCLR. This figure demonstrates where the data augmentaiton (DA) happens in each method and also how the nearest neighbor (NN) search is different between NNCLR and GPS-SimCLR. Note that the 'queue' in NNCLR has a limited size, usually set to 65536. This issue could lead to under-represented classes to not be learned efficiently.

Table 1: Classification accuracy of a ResNet18 in different ablation settings; **Left:** Comparison GPS-SimCLR when different pretrained networks are used for generating embeddings for nearest-neighbor calculation, i.e., prior knowledge. **Right:** Best performance in *StrongAug* setting of SimCLR and GPS-SimCLR given different learning rates (LR).

| **GPS-SimCLR** | FGVCAircraft | | **LR** | FGVCAircraft | |
|---|---|---|---|---|---|
| | *RHFlipAug* | *StrongAug* | | SimCLR | GPS-SimCLR |
| $ViTB_{MAE}$ | 10.53 | 29.55 | 0.003 | 21.39 | 35.7 |
| $ViTL_{MAE}$ | 14.70 | 35.28 | 0.01 | 30.18 | 43.68 |
| $RN50_{SUP}$ | 18.15 | 41.47 | 0.03 | 39.27 | 49.57 |
| $RN50_{VAE}$ | 11.04 | 32.06 | 0.1 | **39.81** | **50.08** |
| $RN50_{CLIP}$ | **19.38** | **50.08** | 0.3 | **39.87** | **48.10** |

In order to thoroughly validate GPS-SSL as an all-purpose strategy for SSL, we consider SimCLR, BYOL, NNCLR, and VICReg as baseline SSL models, and for each of them, we will consider the standard SSL positive pair generation (eq. (2)) and the proposed one (eq. (3)). We opted for a *randomly-initialized* backbone ResNets (He et al., 2016) as the encoder. We also bring forward the fact that most SSL methods are generally trained on a large dataset for which strong DAs are known and well-tuned, such as Imagenet, and the learned representation is then transferred to solve tasks on smaller and less known datasets. In many cases, training those SSL models directly on those atypical dataset lead to catastrophic failures, as the optimal DAs have not yet been discovered. Lastly, we will consider five different embeddings for $g_\gamma$, one obtained from supervised learning, one from CLIP training (Radford et al., 2021) trained on LAION-400M Schuhmann et al. (2021), one for VAE (Kingma & Welling, 2013), and two for MAE (He et al., 2022) all trained on ImageNet and furthermore show our method is more robust to hyper-parameter changes (Table 1). Before delving in our empirical experiments, we emphasize that the supervised $g_\gamma$ is employed in the *RHFlipAug* setting for two reasons. First, it provides what could be thought of as an optimal setting where the class invariants have been learned through the label information. Second, as a mean to demonstrate

Table 2: Classification accuracy of baseline SSL methods with and without GPS-SSL on four datasets on **ResNet50** using pretrained $RN50_{CLIP}$ embeddings for positive sampling. We consider both *StrongAug* (Strong Augmentation) and *RHFlipAug* (Weak Augmentation) settings. The set of DA used for *StrongAug* are `random-resized-crop`, `random-horizontal-flip`, `color-jitter`, `gray-scale`, `gaussian-blur`, and `solarization`. For the *RHFlipAug* setting, the only DA used is `random horizontal flip`. We mark the **first**, second, and third best performing models accordingly.

| Aug. | Method | Datasets | | | |
|------|--------|----------|----------|----------|----------|
| | | Cifar10 (10 classes) | FGVCAircraft (100 classes) | PathMNIST (9 classes) | TissueMNIST (8 classes) |
| *RHFlipAug* | SimCLR | 47.01 | 5.61 | 63.42 | 50.35 |
| | BYOL | 41.79 | 6.63 | 67.08 | 48.00 |
| | NNCLR | 28.46 | 6.33 | 56.70 | 37.98 |
| | Barlow Twins | 41.73 | 5.34 | 53.27 | 43.57 |
| | VICReg | 37.51 | 6.18 | 46.46 | 39.79 |
| | GPS-SimCLR (ours) | 85.08 | 18.18 | 87.79 | 53.14 |
| | GPS-BYOL (ours) | 84.07 | 13.50 | 87.67 | 53.05 |
| | GPS-Barlow (ours) | 84.45 | 17.34 | 88.77 | 56.63 |
| | GPS-VICReg (ours) | 85.58 | 18.81 | 88.91 | 56.44 |
| *StrongAug* | SimCLR | 90.24 | 47.11 | 93.64 | 58.53 |
| | BYOL | 90.50 | 34.23 | 93.29 | 56.63 |
| | NNCLR | 90.03 | 34.80 | 92.87 | 52.57 |
| | Barlow Twins | 88.34 | 18.12 | 92.03 | 61.69 |
| | VICReg | 91.21 | 38.74 | 93.22 | 60.18 |
| | GPS-SimCLR (ours) | 91.17 | 55.60 | 92.30 | 55.59 |
| | GPS-BYOL (ours) | 91.15 | 44.28 | 92.40 | 55.03 |
| | GPS-Barlow (ours) | 88.52 | 15.47 | 91.98 | 57.04 |
| | GPS-VICReg (ours) | 89.71 | 47.29 | 92.55 | 55.79 |

how one could combine a supervised dataset as a mean to produce prior information into training an SSL model on a different dataset. Since the said models are trained on ImageNet, all the provided results throughout this study remain practical since the labels of the target datasets, on which SSL models are trained and evaluated, are never observed for the training of neither $g_\gamma$ nor $f_\theta$.

**Strong Augmentation Experiments.** The DAs in the *StrongAug* configuration consist of strong augmentations that usually distort the size, resolution, and color characteristics of the original image. First, we note that in this setting, GPS-SSL generally does not harm the performance of the baseline SSL model on common datasets, i.e. Cifar10 (Table 2). In fact, GPS-SSL performs comparable to the best-performing baseline SSL model on Cifar10, i.e., VICReg, showcasing that GPS-SSL does not negatively impact performances even on those datasets. We believe that the main reason lies in the fact that the employed DA has been specifically designed for those datasets (and ImageNet). However, we observe that GPS-SSL outperforms (on FGVCAircraft and TissueMNIST) or is comparable to (on PathMNIST) the baseline SSL methods for the under-studied and real-word datasets. (Table 2). The reason for this is that, the optimal set and configuration of DA for one dataset is not necessarily the optimal set and configuration for another, and while SSL solely relies on DA for its positive samples, GPS-SSL is able to alleviate that dependency through $g_\gamma$ and uses positve samples that can be more useful than default DAs, as seen in Figure 2. The results for these experiments can be seen in Table 2. Note that our method's runtime is similar to the baseline SSL method on the dataset it is learning and does not hinder the training process (Figure 4).

**Weak-Augmentation Experiments.** We perform all experiments under the *RHFlipAug* setting as well, showing GPS-SSL produces high quality representations even in that setting, validating theorem 1. As seen in Table 2, GPS-SSL significantly outperforms all baseline SSL methods across both well-studied and under-studied datasets. These results show that our GPS-SSL strategy, though conceptually simple, coupled with the *RHFlipAug* setting, approximates strong augmentations used in the *StrongAug* configuration. This creates a significant advantage for GPS-SSL to be applied to real-world datasets where strong augmentations have not been found, but where the invariances learned by $g_\gamma$ to generalize to them.

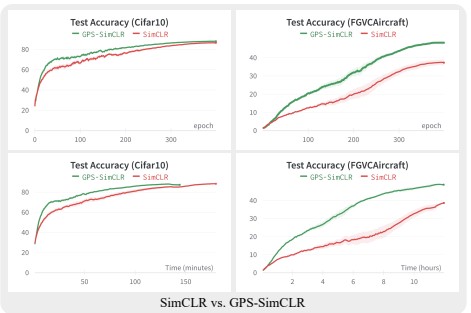 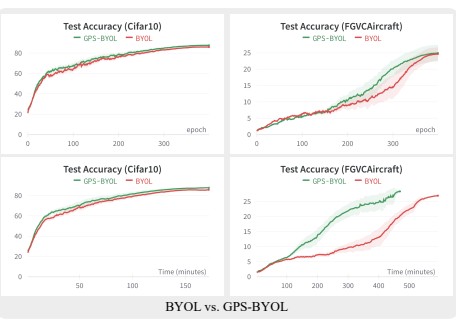

SimCLR vs. GPS-SimCLR    BYOL vs. GPS-BYOL

Figure 4: Comparing the runtime of BYOL vs. GPS-BYOL and SimCLR vs. GPS-SimCLR for two datasets, i.e., FGVCAircraft and Cifar10. In general, we see while the runtime of GPS-SSL remains the same as the original baseline SSL method, it improves the performance.

Table 3: **Test accuracy comparison of GPS-SSL after 1K epochs versus 400 training epochs. We show the improvements of GPS-SimCLR are still significant on FGVCAircraft and comparable on Cifar10.**

| Method | Cifar10 | | FGVCAircraft | |
|---|---|---|---|---|
| | **400 eps** | **1000 eps** | **400 eps** | **1000 eps** |
| **SimCLR** | 88.26 | 91.25 | 39.87 | 45.55 |
| **GPS-SimCLR** | 89.57 | 91.10 | 50.08 | 51.64 |
| **VICReg** | 89.34 | 90.61 | 33.21 | 41.19 |
| **GPS-VICReg** | 89.68 | 89.84 | 45.48 | 49.29 |

**Ablation Study** In this section we explore multiple ablation experiments in order to show GPS-SSL improves SSL and is indeed a future direction for improving SSL methods. **First, we compare SSL and GPS training on Cifar10 and FGVCAircraft starting from a backbone initialized with random (realistic setting), supervised ImageNet pretrained, or CLIP pretrained weights to explore whether the improvement of GPS-SSL is due to better positive sampling or simply because of using a strong prior knowledge. We show in Table 4 that GPS-SSL performs better than the baseline SSL methods, even when they both have access to the pretrained network weights. This proves that the improvement in performance of GPS-SSL compared to baseline SSL methods is indeed due to better positive sampling.**

Next, we compare GPS-SimCLR with three different embeddings for $g_\gamma$; supervised, VAE, and CLIP embeddings. We observe that as the embeddings get higher in quality based on the pre-trained network, as the performance increases in both the *RHFlipAug* and *StrongAug* setting. However, note that even given the worst embeddings, i.e., the VAE embeddings, GPS-SimCLR still outperforms the original SimCLR in the *RHFlipAug* setting, showcasing that the nearest neighbors add value to the learning process when the augmentations are unknown.

**We further explore if the improvement of GPS-SSL holds when methods are trained longer. To that end, we train a ResNet18 for 1000 epochs with SimCLR and VICReg with *StrongAug*, along with their GPS versions, on Cifar10 and FGVCAircraft and compare the results with the performance from 400 epochs. As seen in Table 3, the improvement of GPS-SSL compared to the baseline SSL method holds on FGVCAircraft dataset and remains comparable on Cifar10, showcasing the robustness of GPS-SSL.**

Finally, we aim to measure the sensitivity of the performance of a baseline SSL method to a hyper-parameter, i.e., learning rate, with and without GPS-SSL. In this ablation experiment, we report the best performance of SimCLR and GPS-SimCLR given different learning rates in the *StrongAug* setting. We observe that GPS-SSL when applied to a baseline SSL method is as much, if not more, robust to hyper-parameter change. The results of both ablations are reported in Table 1. We further compare GPS-SSL with linear probing's performance and other ablations in Appendix A.3.

## 4 CASE STUDY ON THE HOTELS IMAGE DATASET

In this section, we study how GPS-SSL compares to baseline SSL methods on an under-studied real-world dataset. We opt the R-HID dataset for our evaluation which gathers hotel images for the purpose of countering human-trafficking. R-HID provides a single train set alongside 4 evaluation sets, each with a different level of difficulty.

Table 4: **Comparing SimCLR with and without GPS-SimCLR with different initializations with a ResNet50. RAND, PT$_{SUP}$, and PT$_{CLIP}$ represent random weights, ImageNet supervised weights, and CLIP pretrained weights.**

| Method | Weight Init. | Cifar10 | | FGVCAircraft | |
|---|---|---|---|---|---|
| | | Weak Aug | Strong Aug | Weak Aug | Strong Aug |
| **SimCLR** | **RAND** | **46.69** | **87.39** | **5.67** | **27.36** |
| **GPS-SimCLR** | | **85.2** | **90.48** | **17.91** | **43.56** |
| **SimCLR** | **PT$_{SUP}$** | **43.99** | **94.02** | **17.91** | **59.92** |
| **GPS-SimCLR** | | **91.3** | **95.53** | **39.45** | **66.88** |
| **SimCLR** | **PT$_{CLIP}$** | **45.57** | **90.26** | **6.21** | **41.04** |
| **GPS-SimCLR** | | **89.44** | **91.23** | **24.15** | **49.63** |

Table 5: R@1 on different spltis on R-HID Dataset for SSL methods. The splits are namely, $\mathcal{D}_{SS}$: {branch: seen, chain: seen}, $\mathcal{D}_{SU}$: {branch: unseen, chain: seen}, $\mathcal{D}_{UU}$: {branch: unseen, chain: unseen} and $\mathcal{D}_{??}$: {branch: unknown, chain: unknown}. We mark the best performing score in **bold**.

| Method | $\mathcal{D}_{SS}$ | $\mathcal{D}_{SU}$ | $\mathcal{D}_{UU}$ | $\mathcal{D}_{??}$ |
|---|---|---|---|---|
| SimCLR | 3.28 | 16.76 | 20.30 | 16.00 |
| BYOL | 3.69 | 19.27 | 23.02 | 18.47 |
| Barlow Twins | 3.04 | 15.54 | 18.96 | 15.06 |
| VICReg | 3.41 | 17.52 | 20.45 | 16.53 |
| GPS-SimCLR (ours) | 4.84 | 23.67 | 26.30 | 22.28 |
| GPS-BYOL (ours) | 3.89 | 19.64 | 23.18 | 19.38 |
| GPS-Barlow (ours) | 4.49 | 21.98 | 25.23 | 20.82 |
| GPS-VICReg (ours) | **5.33** | **25.71** | **28.29** | **23.78** |

We evaluate the baseline SSL models with and without GPS-SSL to the R-HID dataset and report the Recall@1 (R@1) for the different splits introduced. Based on the findings from 2, we adapt the *StrongAug* setting along with the prior knowledge generated by a CLIP-pretrained ResNet50.

As seen in Table 5, SSL baselines always get an improvement when used with GPS-SSL. The reason the baseline SSL methods underperform compared to their GPS-SSL version is that the positive samples generated only using DA lack enough diversity since the images from R-HID dataset have various features and merely DAs limits the information the network learns; however, paired with GPS-SSL, we see a clear boost in performance across all different splits due to the additional information added by the neareset neighbors.

## 5 CONCLUSIONS

In this paper we proposed GPS-SSL which presents a novel strategy to obtain positive samples for Self-Supervised Learning. In particular, GPS-SSL moves away from the usual DA-based positive sampling by instead producing positive samples from the nearest neighbors of the data as measure in some prescribed embedding space. That is, GPS-SSL introduces an entirely novel axis to research and improve SSL that is complementary to the design of DA and losses. Through that strategy, we were for example able to train SSL on atypical datasets such as medical images –without having to search and tune for the right DA. Those results open new avenues to the existing strategy of SSL pretraining on large dataset, and then transferring the model to those other datasets for which DAs as not available. In fact, we observe that while GPS-SSL meets or surpass SSL performances across our experiments, the performance gap is more significant when the optimal DAs are not known, e.g., in PathMNIST and TissueMNIST we observe the performance of GPS-SSL with weak augmentations is slightly less than with strong augmentations. Besides practical applications, GPS-SSL finally provides a novel strategy to embed prior knowledge into SSL.

**Limitations.** The main limitation of our method is akin to the one of SSL, it requires the knowledge of the embedding in which the nearest neighbors are obtain to produce the positive samples. This limitation is on par with standard SSL's reliance on DA, but its formulation is somewhat dual (recall theorem 1) in that one may know how to design such an embedding without knowing the appropriate DA for the dataset, and vice-versa. Alternative techniques like training separate and simple DNs to provide such embeddings prior to the SSL learning could be considered for future research.

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

## A  APPENDIX

### A.1  R-HID SPLITTING METHOD

**R-HID (Feizi et al., 2022) is created carefully to make sure no data leakage occurs. They mention how the total data is divided into the train and the multiple test splits. More specifically, first a set of chains (along with *all* their branches) are reserved for the $\mathcal{D}_{UU}$ to make sure the chains (super-classes) and branches (classes) are not seen during training. Next, out of the remaining chains, a set of the branches are chosen to add *all* of their images to the $\mathcal{D}_{SU}$ test split (since the training set will have other images from other branches from the same chain, but not the same branch images). Finally, out of the remaining branches, the images in each are split between $\mathcal{D}_{SS}$ and train, creating the final test split that has a subset of the branches seen during training. With this procedure, they make sure of the table of overlapping below. More details regarding the splits is provided in the original paper.**

### A.2  HYPER-PARAMETER SEARCH

In all experiments, we train for **400 epochs** with a batch size of 256 using one RTX 8000 GPU for all methods. To ensure we are choosing the correct hyper-parameters for a fair comparison, we search over a vast range of hyper-parameter combinations ($lr \in \{1e^{-3}, 3e^{-3}, 3e^{-2}, 1e^{-2}, 3e^{-1}, 1e^{-1}, 1\}$, $classifier\_lr \in \{3e^{-2}, 1e^{-2}, 3e^{-1}, 1e^{-1}, 1, 3\}$, $weight\_decay \in \{1e^{-4}, 1e^{-3}\}$) and for GPS-SSL with all SSL baselines we also search over $k \in \{1, 4, 9, 49\}$). **For experiments using *RHFlipAug* and *StrongAug*, we use nearest neighbors calculated based on embeddings created from a ResNet50 that have been CLIP pre-trained as the prior knowledge.** Finally, for each method, we report the best classification accuracy for Cifar10, FGVCAircraft, PathMNIST, and TissueMNIST, and Recall@1 (R@1) for R-HID in Tables 2, 5, and 7. To calculate both metrics, we first train the encoder on the target dataset using the SSL method, with or without GPS-SSL. Then, for classification accuracy, we train a linear classifier on top of it, and for R@1, we encode all the images from the test set and calculate the percentage of images which their first nearest neighbor is from the same class.

### A.3  ABLATION STUDY

#### A.3.1  DIFFERENT BACKBONE

**First, we provide the same experiments as in Table 2, but trained with a ResNet18 instead of a ResNet50 and provide the results in Table 6. We see the same results for ResNet50 (discussed for Table 2) also hold when ran on a smaller architecture, i.e., ResNet18. This shows the improvements of GPS-SSL over baseline SSL methods is more reliable and robust.**

#### A.3.2  FINETUNING FOR R-HID

We further try a trivial way of transferring knowledge from a pretrained network to other SSL baseline models and compare it to GPS-SimCLR; we initialize the base encoder in any SSL method, i.e., the ResNet18, to the pretrained network's weights, as opposed to random initialization, and train it i.e., finetuning. Ultimately, we compare the results on R-HID in Table 7.

Although this might perform better if the pretrained network was trained on a visually similar dataset to the target dataset, Table 7 shows that it may harm the generalization on datasets that are different, e.g., ImageNet and R-HID, compared to being trained from scratch. However, GPS-SSL proves to

Table 6: Classification accuracy of baseline SSL methods with and without GPS-SSL on four datasets on *ResNet18* using pretrained $RN50_{CLIP}$ embeddings for positive sampling.. We consider both *StrongAug* (Strong Augmentation) and *RHFlipAug* (Weak Augmentation) settings. The set of DA used for *StrongAug* are `random-resized-crop`, `random-horizontal-flip`, `color-jitter`, `gray-scale`, `gaussian-blur`, and `solarization`. For the *RHFlipAug* setting, the only DA used is `random horizontal flip`. We mark the **first**, second, and third best performing models accordingly.

| Aug. | Method | Datasets | | | |
|---|---|---|---|---|---|
| | | Cifar10 (10 classes) | FGVCAircraft (100 classes) | PathMNIST (9 classes) | TissueMNIST (8 classes) |
| *RHFlipAug* | SimCLR | **47.62** | **7.70** | **62.99** | **52.30** |
| | BYOL | **49.72** | **8.99** | **77.77** | **51.00** |
| | NNCLR | **71.74** | **8.10** | **56.92** | **42.59** |
| | Barlow Twins | **42.00** | **7.53** | **64.82** | **49.43** |
| | VICReg | **36.04** | **4.95** | **56.92** | **50.26** |
| | GPS-SimCLR (ours) | **85.83** | **18.48** | **88.62** | **55.98** |
| | GPS-BYOL (ours) | **84.56** | **14.79** | **81.66** | **56.21** |
| | GPS-Barlow (ours) | **84.83** | **18.12** | **87.79** | **55.86** |
| | GPS-VICReg (ours) | **85.38** | **20.16** | **87.83** | **55.26** |
| *StrongAug* | SimCLR | 88.26 | 39.87 | 91.56 | 61.51 |
| | BYOL | 86.90 | 27.33 | 91.24 | 60.73 |
| | NNCLR | 87.95 | 39.12 | 91.14 | 52.42 |
| | Barlow Twins | 88.89 | 25.71 | 92.23 | 60.06 |
| | VICReg | 89.34 | 33.21 | 92.27 | 59.41 |
| | GPS-SimCLR (ours) | 89.57 | 50.08 | 92.19 | 62.76 |
| | GPS-BYOL (ours) | 88.46 | 32.07 | 91.05 | 54.05 |
| | GPS-Barlow (ours) | 88.39 | 25.35 | 91.55 | 62.93 |
| | GPS-VICReg (ours) | 89.68 | 45.48 | 91.88 | 62.46 |

be a stable method for transferring knowledge even if the pretrained and target dataset are visually different (Table 5).

Table 7: Comparing the R@1 performance of SSL methods on R-HID when trained from scratch against being initialzied to a ImageNet pretrained network. The models with pretrained-initialized encoders (finetuned) are marked with 'FT-'. We highlight the difference in R@1 of the pretrained against the scratch version with green when it improves and red when it worsens.

| Method | $\mathcal{D}_{SS}$ | $\mathcal{D}_{SU}$ | $\mathcal{D}_{UU}$ | $\mathcal{D}_{??}$ |
|---|---|---|---|---|
| SimCLR | 3.23 | 16.10 | 19.62 | 15.12 |
| FT-SimCLR | -0.10 | -0.21 | -0.40 | +0.27 |
| BYOL | 3.27 | 16.25 | 20.20 | 15.91 |
| FT-BYOL | -0.57 | -1.75 | -2.23 | -1.50 |
| NNCLR | 2.84 | 13.91 | 17.15 | 13.96 |
| FT-NNCLR | -0.54 | -2.44 | -3.18 | -2.67 |
| VICReg | 3.24 | 16.67 | 19.97 | 15.86 |
| FT-VICReg | -0.43 | -1.54 | -2.45 | -1.92 |

Table 8: **Comparison of Linear probing (LP) and GPS-VICReg's (with ResNet50) classification accuracy on FGVCAircraft with different GPS backbones (GPS BB) pretrained with CLIP and masked auto encoders (MAE) on different datasets without supervision (GPS DS). The performance of the vanilla VICReg is also depicted for comparison. RN50 and ViT-L refer to ResNet50 and ViT-Large, respectively.**

| GPS-BB | GPS-DS | LP | GPS-VICReg | VICReg |
|--------|--------|-----|-----------|--------|
| $RN50_{CLIP}$ | LAION-400M | 44.55 | 46.44 | |
| $ViT\text{-}L_{MAE}$ | ImageNet | 37.32 | 38.44 | 39.99 |
| $ViT\text{-}L_{MAE}$ | FGVCAircraft | 17.01 | 42.87 | |

Table 9: **Classification accuracy comparison of linear probing (LP) using embeddings with different GPS backbones (GPS-BB) pretrained with CLIP and masked auto encoders (MAE) on different upstream datasets, i.e. GPS-DS, and a trained ResNet50 with GPS-SimCLR on FGVCAircraft and Cifar10 using the same GPS backbones and datasets. RN50, ViT-L, and Vit-B refer to ResNet50, ViT-Large, and ViT-Base respectively.**

| GPS-BB | GPS-DS | Cifar10 | | FGVCAircraft | |
|--------|--------|---------|------------|--------------|------------|
| | | LP | GPS-SimCLR | LP | GPS-SimCLR |
| $RN50_{CLIP}$ | LAION-400M | 87.85 | 91.17 | 44.55 | 53.81 |
| $ViT\text{-}B_{MAE}$ | ImageNet | 85.78 | 87.35 | 27.96 | 29.55 |
| $ViT\text{-}L_{MAE}$ | ImageNet | 91.45 | 90.11 | 37.29 | 35.28 |
| $ViT\text{-}L_{MAE}$ | FGVCAircraft | —— | —— | 17.01 | 46.93 |

### A.3.3 COMPARING TO LINEAR PROBING

**Finally, we compare the linear probing performance of the embeddings generated from different architectures, i.e. GPS backbones (GPS-BB), pretrained on different datasets, i.e., GPS Datasets (GPS-DS), with the performance of GPS-SSL using them. More specifically, in Tables 9 and 8, we compare the linear probe performance of the CLIP pretrained ResNet50 on LAION-400M (Schuhmann et al., 2021) along with vision transformers (ViTs) pretrained on ImageNet using Masked Auto Encoders (MAE) (He et al., 2022), a popular self-supervised method that also does not rely on strong augmentations. We see our method outperforms the linear probe accuracy of CLIP embeddings for both Cifar10 and FGVCAircraft and matches that of ViT-Base and ViT-Large for Cifar10 and ViT-Large for FGVCAircraft.**

**However, we further see that if we train the ViT-Large on the FGVCAircraft, using MAE with minimal augmentations, we can use that as the positive sampler for GPS-SSL and beat the baseline SSL method on FGVCAircraft. This shows that GPS-SSL does not entirely rely on huge pretrained models and that there is potential possibilities for training a positive sampler prior to applying GPS-SSL to further boost the performance of baseline SSL methods.**

