# OpenReview forum: "GPS-SSL: Guided Positive Sampling to Inject Prior into Self-Supervised Learning"
_ICLR.cc/2024/Conference — Submitted to ICLR 2024_

### Official Review · Reviewer_sJCd · 2023-10-26

**Soundness:** 2 fair
**Presentation:** 2 fair
**Contribution:** 2 fair
**Rating:** 5
**Confidence:** 4

**Summary:**

A method using prior knowledge to sample the positive data is proposed. It is supposed to mitigate the importance of data augmentation in self-supervised learning. The proposed GPS-SSL has shown superior capability over the methods with existing augmentation strategies.

**Strengths:**

+ Studying new strategies that rely less on data augmentations in self-supervised learning is worthwhile to the representations learning fields.
+ Exploring the pre-trained models (CLIP, Supervised models, VAE) for improving SSL might be interesting.

**Weaknesses:**

+ The proposed method needs a heavier component (such as a neural network ResNet-50) to generate the positive data sample, which is significantly computational compared to a simple calculation of data augmentation even for strong augmentations with a series of cropping, color jittering, distortion, hue, etc...

+ With the aid of a strong knowledge (and heavy) model trained on millions or hundred million of data (CLIP, ImageNet) the performance of the proposed method brings minimal advantage even worse than the existing SSL method such as VICReg in Table 2 with strong augmentation. In the weak augmentation setting, GPS-SSL may give better performance but still lag significantly behind the optimal setting (strong augmentation) of both streams, making it questionable about the contribution of the proposed method.

+ SSL contains another branch that is also very promising with the fine-tuning accuracy on downstream tasks such as MAE [1], this approach also depends very little on data augmentation (only cropping or without any augmentation already made the very good performance). This example (MAE method) will challenge the proposed method in terms of dependency on augmentation because the proposed method could not work without augmentation. I believe that modern SSLs should include this metric (fine-tune accuracy) and compare both contrastive learning and MAE approaches.

+ It should also include the linear evaluation of the only CLIP RN50 or supervised RN50 model when they have been used as the feature extractor for the downstream tasks on each considered dataset. It is to see without any training, how well these pre-trained model can perform, and based on that we can assess their contribution to the GPS-SSL (which is a combination of existing SSL + pre-trained CLIP/RN50).

+ Another point is that the experimental setting is not practical and sufficient to demonstrate the effectiveness of GPS-SSL when evaluating self-supervised contrastive learn is that they only consider pretraining with 200 epochs, which is very few epochs required by SSL models to fully converge. As shown in SimSiam or many SSL (MoCo, BYOL, Barlow Twins, VICREG,... ) the performance is best achieved with long enough self-supervised pretraining (800-1000 epochs). As a result, the comparison in long training should be considered for both methods.

+ It is not clear what is the metric they have shown in Table 1. Reading its caption, it is challenging to capture what metric they are comparing, top-1 ACC or error or something else.

[1] Masked Autoencoders Are Scalable Vision Learners, CVPR 2022

**Questions:**

See weaknesses

---

> ### Author Response · Authors · 2023-11-17
> **Part 1/3**
>
> - **The proposed method needs a heavier component (such as a neural network ResNet-50) to generate the positive data sample, which is significantly computational compared to a simple calculation of data augmentation even for strong augmentations with a series of cropping, color jittering, distortion, hue, etc…**
>
> *Response:* Very good point; thank you for raising it. First of all, we will make sure to mention that explicitly in the introduction and when we derive our method. That being said, while the addition of another DN is clearly a more intense computational burden, we show that our method converges faster (thus needs fewer epochs, compensating for the extra cost). For example, we now provide a new experiment with a longer training schedule and obtain matching or improved results on both datasets with GPS-SimCLR and GPS-VICReg.
>
> |            | Cifar10 |          | FGVCAircraft |          |
> |------------|:-------:|:--------:|:------------:|:--------:|
> |            | 400 eps | 1000 eps |    400 eps   | 1000 eps |
> |   SimCLR   |  88.26  |   91.25  |     39.87    |   45.55  |
> | GPS-SimCLR |  89.57  |   91.10  |     50.08    |   51.64  |
> |   VICReg   |  89.34  |   90.61  |     33.21    |   41.19  |
> | GPS-VICReg |  89.68  |   89.84  |     45.48    |   49.29  |
>
>
>
> Also, it is possible that the NN sampling, relying only on the latent space representations, can be made on a saved bank of representations obtained from the NN architecture, i.e., no need to forward each iteration. We will mention that in the conclusion.
>
> - **With the aid of a strong knowledge (and heavy) model trained on millions or hundred million of data (CLIP, ImageNet) the performance of the proposed method brings minimal advantage even worse than the existing SSL method such as VICReg in Table 2 with strong augmentation. In the weak augmentation setting, GPS-SSL may give better performance but still lag significantly behind the optimal setting (strong augmentation) of both streams, making it questionable about the contribution of the proposed method.**
>
> *Response:* We thank the reviewer for raising that concern. First, we would like to mention that one important contribution of our submission is in demonstrating a new path for SSL methods: improving the positive pair sampling strategy. To that end, we did employ, in a few places, some pretrained embedding that had access to more samples than the SSL baseline (although this was already clearly stated in our manuscript when such results were reported). However, we completely understand that this may limit the applicability and reach of our claims. As such, we have proposed two new sets of experiments to address that concern. The first is to actually let the SSL baseline benefit from that same additional information by replacing the randomly initialized backbone with the same one being used by GPS’s embedding. In short, all methods now have access to the exact same extra information from the get-go:
>
>
> |            |              |  Cifar10 |            | FGVCAircraft |            |   |
> |------------|--------------|:--------:|:----------:|:------------:|:----------:|---|
> |            |              | Weak aug | Strong Aug | Weak aug     | Strong Aug |
> |   SimCLR   | RN50         |    46.69 |      87.39 |         5.67 |      27.36 |
> | GPS-SimCLR |              |     85.2 |      90.48 |        17.91 |      43.56 |
> |   SimCLR   | RN50_pt      |    43.99 |      94.02 |        17.91 |      59.92 |
> | GPS-SimCLR |              |     91.3 |      95.53 |        39.45 |      66.88 |
> |   SimCLR   | RN50_CLIP_pt |    45.57 |      90.26 |         6.21 |      41.04 |
> | GPS-SimCLR |              |    89.44 |      91.23 |        24.15 |      49.63 |
>
> We observe that even in this setting, GPS is able to consistently improve the final SSL representations compared to the baseline (more details in our general answer). The second experiment we propose is to employ GPS with an embedding model that is trained on the same dataset being considered, i.e., no additional knowledge is present in GPS training:
>
> |            |    dataset   | prior knowledge BB | NN_trained_on | backbone | val_acc1 |
> |------------|:------------:|:------------------:|:-------------:|:--------:|:--------:|
> |   VICReg   | FGVCAircraft |          -         |       -       | ResNet50 |   39.99  |
> | GPS-VICReg | FGVCAircraft |      MAE_vitL      |  FGVCAircraft | ResNet50 |   42.87  |
> |   SimCLR   | FGVCAircraft |          -         |       -       | ResNet50 |   47.11  |
> | GPS-SimCLR | FGVCAircraft |      MAE_vitL      |  FGVCAircraft | ResNet50 |   46.93  |
>
>
>
> Where we observe that GPS is again able to match or improve the SSL baseline.

---

> ### Author Response · Authors · 2023-11-17
> **Part 2/3**
>
> - **SSL contains another branch that is also very promising with the fine-tuning accuracy on downstream tasks such as MAE [1], this approach also depends very little on data augmentation (only cropping or without any augmentation already made the very good performance). This example (MAE method) will challenge the proposed method in terms of dependency on augmentation because the proposed method could not work without augmentation. I believe that modern SSLs should include this metric (fine-tune accuracy) and compare both contrastive learning and MAE approaches.**
>
> *Response:* We thank the reviewer for this suggestion. Indeed, once allowing for fine-tuning, we believe that most of those variants should produce comparable performances. However, we focused on a frozen backbone as a means to precisely assess the quality of the learned representations, as guided by the employed training criterion. In fact, as part of our motivation is to argue on the current sub-optimal positive pair sampling strategy, we resorted to assessing such a performance gap without fine-tuning. We will, however, clearly state this limitation and also mention in our contributions that our goal is to improve frozen backbone SSL.
>
>
> - **It should also include the linear evaluation of the only CLIP RN50 or supervised RN50 model when they have been used as the feature extractor for the downstream tasks on each considered dataset. It is to see without any training, how well these pre-trained model can perform, and based on that we can assess their contribution to the GPS-SSL (which is a combination of existing SSL + pre-trained CLIP/RN50).**
>
> *Response:* That is an excellent point and we thank you for mentioning it. We added a table that depicts the linear probe evaluation of a pretrained RN50 with CLIP on Cifar10 and FGVCAircraft dataset:
>
>
> |                     | Cifar10	 | FGVCAircraft |
> |---------------------|:-------:|:------------:|
> | CLIP_RN50_LAION400M |   0.88  |     0.45     |
> |  MAE_vitB_ImageNet  |   0.86  |     0.28     |
> |  MAE_vitL_ImageNet  |   0.91  |     0.37     |
> | vae_RN50_objects365 |   0.29  |     0.02     |
>
>
> As it can be seen, GPS-SSL indeed improves the classification performance on both datasets when trained on a RN50 with strong augmentations, showing that GPS-SSL indeed contributes. Furthermore, we show that even when using a CLIP pretrained RN50, GPS-SimCLR outperforms SimCLR in all states, showing that the contribution is due to the better “positive sampling”, rather than using a pre-trained RN50.
>
> |           |    cifar10	   |            | FGVCAircraft |            |
> |-----------|:------------:|:----------:|:------------:|:----------:|
> |           | linear probe | GPS-SimCLR | linear probe | GPS-SimCLR |
> | CLIP_RN50 |     87.85    |    91.17   |     44.55    | 53.81      |

---

> ### Author Response · Authors · 2023-11-17
> **Part 3/3**
>
> - **Another point is that the experimental setting is not practical and sufficient to demonstrate the effectiveness of GPS-SSL when evaluating self-supervised contrastive learn is that they only consider pretraining with 200 epochs, which is very few epochs required by SSL models to fully converge. As shown in SimSiam or many SSL (MoCo, BYOL, Barlow Twins, VICREG,... ) the performance is best achieved with long enough self-supervised pretraining (800-1000 epochs). As a result, the comparison in long training should be considered for both methods.**
>
> *Response:* We thank the reviewer for raising this limitation, we entirely agree with the need to provide a longer training schedule. We had previously reported results where most of the models were trained for 400 epochs, and some were trained for 200 epochs. The first change we have conducted following your suggestion is to make all the models in the reported tables in the submission with 400 training epochs. (Table 2 and 6).
>
> Second, we performed a few additional experiments training for 1000 epochs with a ResNet18 using SimCLR (with and without GPS) which we report below (and in Table 3)
>
> |            | Cifar10 |          | FGVCAircraft |          |
> |------------|:-------:|:--------:|:------------:|:--------:|
> |            | 400 eps | 1000 eps |    400 eps   | 1000 eps |
> |   SimCLR   |  88.26  |   91.25  |     39.87    |   45.55  |
> | GPS-SimCLR |  89.57  |   91.10  |     50.08    |   51.64  |
> |   VICReg   |  89.34  |   90.61  |     33.21    |   41.19  |
> | GPS-VICReg |  89.68  |   89.84  |     45.48    |   49.29  |
>
> Where we employ strong augmentations (the standard SSL setting). We observe that GPS consistently improves performances and arrives on par with SimCLR on Cifar10 after 1000 epochs. Interestingly (and we thank the reviewer again for the suggestion that led to quantifying that benefit) we observe that GPS converges in fewer epochs than SSL. This is again another possible benefit of opening new avenues for informed positive pair sampling: faster SSL training. We hope that those novel experiments answered the reviewer’s empirical validation concerns.
>
> - **It is not clear what is the metric they have shown in Table 1. Reading its caption, it is challenging to capture what metric they are comparing, top-1 ACC or error or something else.**
>
> *Response:* We apologize for the confusion, the reported metrics are top-1 accuracy, we have made sure to state that clearly in the caption, thank you for raising that discrepancy.

---

> ### Author Response · Authors · 2023-11-21
> **Reminder**
>
> Dear Reviewer sJCd,
>
> We hope that you have had a chance to consider our rebuttal. Your comments greatly helped us improve our manuscript, we sincerely thank you for that. We would like to kindly remind you that ICLR won't have a second round of discussion. As such, we would deeply appreciate it if you could raise any remaining concern or comment that you may have. This way, we will have a chance to clarify any last issue before the end of the discussion period which is now very near. If you consider your concerns addressed, then we hope that you will find time to revise your score and review accordingly.
> Thank you again for participating in the peer-review process.
>
> Best regards,
> The authors

---

> ### Comment · Reviewer_sJCd · 2023-11-22
>
> Thanks authors for their response. The rebuttal partly addressed my concerns. However, the additional results remain that the proposed method still lags behind the traditional SSL with the aid of the strong knowledge of the prior CLIP model (as mentioned in the weakness before).
> + For example, in CIFAR-10 with optimal training (1000 ep), it clearly shows that SimCLR and VIGReg are still better than the proposed method (meaning that GPS failed to improve the SSL baselines in the optimal setting).
> + For the FGVCAircraft dataset, although it can show some improvements, I think it is a very weak baseline, for example, paper [1] (Table 2) shows that SSL baselines such as MoCo-2 already achieved 52.54\% and [1] even achieves 55.87\% which is by far better than the results provided by GPS.
>
> [1] Learning Common Rationale to Improve Self-Supervised Representation for Fine-Grained Visual Recognition Problems, CVPR 2023
>
> I tend to keep my initial rating.

---

> > ### Author Response · Authors · 2023-11-22
> > **Response**
> >
> > Dear Reviewer sJCd,
> >
> > We want to thank you for acknowledging that our rebuttal answered most of your comments, and for raising two last concerns:
> >
> > 1. We would like to argue that such an optimal setting (CIFAR being the dataset where SSL methods have been designed and engineered for alongside ImageNet) was provided as a sanity check to ensure that the proposed method was able to recover that upper bound. As you clearly stated in your answer, this should be considered **the optimal setting for SSL**. We thus do not see this point as a reason for rejection and to disregard the many other contributions and improved performances we obtained.
> > 2. We are greatly appreciative of the reviewer providing an external benchmark result. After reading through that paper we noticed that the reported results (including the one you raised in your answer) are obtained from an ImageNet pretrained backbone as stated in Sec 4.1, page 5 of [1]. The results we reported in Table 2 of our paper on FGVCAircraft is for a *randomly initialized* backbone. If we put ourselves in the same setting (ImageNet ResNet50 initialization and run for 100 epochs), GPS is able to significantly outperform their result as we reach 65.20% with 100 epochs (and even 63.07% with 50 epochs) while they reach 52.54% for MoCo-2 and 55.87% for [1] (similar results are reported in Table 4 in the second row of our paper). This is however an interesting benchmark and we will make sure to reference it and compare against it in our final submission, we will make sure to emphasize the backbone being used in each experiment to avoid confusion.
> >
> > We hope that the reviewer will have the time to consider our answer before the end of the discussion period.
> >
> > Best regards,
> >
> > The authors
> >
> >
> >
> > [1] Learning Common Rationale to Improve Self-Supervised Representation for Fine-Grained Visual Recognition Problems, CVPR 2023

---

### Official Review · Reviewer_uQnq · 2023-10-31

**Soundness:** 3 good
**Presentation:** 3 good
**Contribution:** 2 fair
**Rating:** 6
**Confidence:** 3

**Summary:**

The authors proposed the Guided Positive Sampling (GPS) approach to
finding positive pairs in self-supervised learning, without data
augmentation.  For each instance, a nearest neighbor is found in an
embedding space pretrained with another dataset or with a variational
autoencoder on the same dataset.  The corresponding instance becomes
the positive instance for self-supervised learning.

In their experiments, they consider using GPS with SIMCLR, BYOL,
Barlow, and VICreg on five datasets.  For GPS, they use embeddings
from supervised training, CLIP or VAE.  Generally, empirical
results indicate that using GPS outperforms, particularly with weak
augmentations.

**Strengths:**

Not relying on heavy handcrafting of data augmentation for
self-supervised learning is interesting.  Using prior knowledge based
on a pretrained encoder, they propose to find a nearest neighbor to
form a positive pair.  Generally, empirical results indicate that using
GPS outperforms, particularly with weak augmentations.

**Weaknesses:**

With prior knowledge, GPS seems to have an advantage over regular SSL,
which generally does not use prior knowledge.  According to Figure 1,
data augmentation is used in GPS-SimCLR.  So GPS seems to differ only
in the use of prior knowledge to find positive pairs.

Details are in questions below.

**Questions:**

1.  Theorem 1: GPS-SSL: employing eq (2) or (3) into eq (1)?

2.  Table 2: why are two different kinds of prior knowledge is used?

3.  How is $Tau$ set in Equation 3?

4.  With prior knowledge from another encoder, GPS has an advantage.
    Hence, comparison with methods that don't have prior knowledge
    might not be fair.  Could the regular SSL (with augmentation) also
    use prior knowledge?  For example, the encoder is initialized by
    prior knowledge and then regular SSL is performed.

5.  Sec 4.1, how do you predict if the classes do not overlap in the
    training and test sets (unseen classes branches/chains)?

--------  after response from authors ---

I think the authors performed experiments that remove the advantage of prior knowledge used in GPS and the results indicate GPS can improve performance over regular SSL.

---

> ### Author Response · Authors · 2023-11-17
> **Part 1/2**
>
> -  **With prior knowledge, GPS seems to have an advantage over regular SSL, which generally does not use prior knowledge. According to Figure 1, data augmentation is used in GPS-SimCLR. So GPS seems to differ only in the use of prior knowledge to find positive pairs.**
>
> *Response:* We thank the reviewer for raising that concern (which was shared with other reviewers). We have provided two novel sets of experiments to demonstrate that GPS is able to help SSL learn richer representations by alleviating the sub-optimal positive pair generation employed by SSL that goes far beyond the use of additional knowledge that exists in the pretrained embedding. The first is to actually let the SSL baseline benefit from that same additional information by replacing the randomly initialized backbone with the same one being used by GPS’s embedding. In short, all methods now have access to the exact same extra information from the get-go:
>
> |            |              |  Cifar10 |            | FGVCAircraft |            |
> |------------|--------------|:--------:|:----------:|:------------:|:----------:|
> |            |              | Weak aug | Strong Aug | Weak aug     | Strong Aug |
> |   SimCLR   | RN50         |    46.69 |      87.39 |         5.67 |      27.36 |
> | GPS-SimCLR |              |     85.2 |      90.48 |        17.91 |      43.56 |
> |   SimCLR   | RN50_sup_pt  |    43.99 |      94.02 |        17.91 |      59.92 |
> | GPS-SimCLR |              |     91.3 |      95.53 |        39.45 |      66.88 |
> |   SimCLR   | RN50_CLIP_pt |    45.57 |      90.26 |         6.21 |      41.04 |
> | GPS-SimCLR |              |    89.44 |      91.23 |        24.15 |      49.63 |
>
>
>
> We observe that even in this setting, GPS is able to consistently improve the final SSL representations compared to the baseline (more details in our general answer). The second experiment we propose is to employ GPS with an embedding model that is trained on the same dataset being considered, i.e., no additional knowledge is present in GPS training:
>
> |            |    Dataset   | GPS-Backbone |  GPS-Dataset | Backbone | Val_acc1 |
> |------------|:------------:|:------------:|:------------:|:--------:|:--------:|
> |   VICReg   | FGVCAircraft |       -      |       -      | ResNet50 |   39.99  |
> | GPS-VICReg | FGVCAircraft |   MAE_vitL   | FGVCAircraft | ResNet50 |   42.87  |
> |   SimCLR   | FGVCAircraft |       -      |       -      | ResNet50 |   47.11  |
> | GPS-SimCLR | FGVCAircraft |   MAE_vitL   | FGVCAircraft | ResNet50 |   46.93  |
>
> Where we observe that GPS is again able to match or improve the SSL baseline.
>
> - **Theorem 1: GPS-SSL: employing eq (2) or (3) into eq (1)?**
>
> *Response:* Eq (2) is the standard SSL solution to take a single sample, apply two different data-augmentations to it, and form a positive pair this way. Eq (3) is the proposed strategy obtained by applying the data-augmentation on two samples that are nearest neighbors in some given embedding space. Note that (3) can recover (2) in the limit case where we pick the sample itself as its nearest neighbor, we have put ``GPS’’ as part of eq (3) to make sure that the distinction was as clear as possible.
>
> - **Table 2: why are two different kinds of prior knowledge is used?**
>
> *Response:* We thank the reviewer for raising this concern. We have updated Table 2 accordingly to only use one prior knowledge (RN50_{CLIP} embeddings) for consistency and staying purely in the realm of SSL methods. In short, all employ GPS-SimCLR, the left table shows comparison using GPS with a few different embedding models, while on the right we use GPS-SimCLR with a few different hyper-parameters (to show its stability). All cases are on the same FGVCAircraft dataset.
>
> - **How is \tau set in Equation 3?**
>
> *Response:* That is a very important point, and we thank the reviewer for raising it. Based on the quality of the metric space g_{\gamma}, we can set \tau to be larger or smaller. For example, if we know g_{\gamma} is reliable, we can set \tau to larger values. However, if we are unsure about the reliability of g_{\gamma}, we can set \tau to smaller values to make sure that the positive pairs convey useful information rather than noise. In fact (as mentioned in our original submission) as tau gets close tk 0 as GPS recovers vanilla SSL positive pair sampling (we made sure to emphasize that observation which enables a qualitative guess for the value of \tau.

---

> ### Author Response · Authors · 2023-11-17
> **Part 2/2**
>
> - **With prior knowledge from another encoder, GPS has an advantage. Hence, comparison with methods that don't have prior knowledge might not be fair. Could the regular SSL (with augmentation) also use prior knowledge? For example, the encoder is initialized by prior knowledge and then regular SSL is performed.**
>
> *Response:* We thank the reviewer for raising that concern. Thanks to your suggestion, we have provided that exact experiment in response 1 above (and in the general comment). We believe that this experiment has been extremely insightful to use as it clearly highlights that the current SSL positive pair sampling strategy (eq (2)) is indeed suboptimal, as it moves away from a potentially better-initialized model (as opposed to GPS).
>
> - **Sec 4.1, how do you predict if the classes do not overlap in the training and test sets (unseen classes branches/chains)?**
>
> *Response:* Thank you for raising this concern. The paper that introduces the dataset mentions how they divide the total data into the train and the multiple test splits. More specifically, they explain that first, a set of chains (along with all their branches) is reserved for the D_{UU} to make sure the chains (super-classes) and branches (classes) are not seen during training. Next, out of the remaining chains, a set of branches is chosen to add all of their images to the D_{SU} test split (since the training set will have other images from other branches from the same chain, but not the same branch images). Finally, out of the remaining branches, the images in each are split between D_{SS} and train, creating the final test split that has a subset of the branches seen during training. With this procedure, they ensure the table of overlapping below:
>
> |           | D_{SS}                                                     | D_{SU}                                                   | D_{UU}                                 |
> |-----------|------------------------------------------------------------|----------------------------------------------------------|----------------------------------------|
> | Train set | Shared branches (classes)and shared chains (super-classes) | Only shared chains (super-classes) branches are distinct | Both chains and branches are distinct. |

---

> > ### Comment · Reviewer_uQnq · 2023-11-22
> > **comments on response**
> >
> > Thanks for the response.
> >
> > I think the authors performed experiments that remove the advantage of prior knowledge used in GPS and the results indicate GPS can improve performance over regular SSL.   I plan to upgrade my score.

---

> > > ### Author Response · Authors · 2023-11-22
> > >
> > > Dear Reviewer uQnq,
> > >
> > > We are delighted to read that our rebuttal successfully addressed your concerns. We want to thank you again for responding and for taking the time to do so during the discussion period.
> > >
> > > We remain available until the end of the discussion period to provide any further details or empirical validations that would further reinforce your recommendation for acceptance.
> > >
> > > Best regards,
> > > The authors

---

> ### Author Response · Authors · 2023-11-21
> **Reminder**
>
> Dear Reviewer uQnq,
>
> We hope that you have had a chance to consider our rebuttal. Your comments greatly helped us improve our manuscript, we sincerely thank you for that. We would like to kindly remind you that ICLR won't have a second round of discussion. As such, we would deeply appreciate it if you could raise any remaining concern or comment that you may have. This way, we will have a chance to clarify any last issue before the end of the discussion period which is now very near. If you consider your concerns addressed, then we hope that you will find time to revise your score and review accordingly.
> Thank you again for participating in the peer-review process.
>
> Best regards,
> The authors

---

### Official Review · Reviewer_aLRr · 2023-11-01

**Soundness:** 3 good
**Presentation:** 3 good
**Contribution:** 2 fair
**Rating:** 5
**Confidence:** 3

**Summary:**

The paper proposes Guided Positive Sampling Self-Supervised Learning (GPS-SSL), a method that integrates prior knowledge into Self-Supervised Learning (SSL) to improve positive sample selection and reduce reliance on data augmentations. Based on pretrained visual models and target dataset, GPS-SSL creates a metric space that facilitates nearest-neighbor sampling for positive samples. The method is applicable to various SSL techniques and outperforms baseline methods, particularly when minimal augmentations are used.

**Strengths:**

- Extensive experiments show the effectiveness of the GPS strategy.
- The paper is easy to follow.

**Weaknesses:**

- The employment of prior knowledge, specifically in the form of a pretrained visual model and the target dataset, diverges from the fundamental principles of Self-Supervised Learning (SSL).
- The incorporation of such prior knowledge raises concerns about the fairness of comparisons with existing SSL methods. There is a potential risk that the pretrained visual model and target dataset might leak additional information into the model, thereby skewing results and leading to issues of unfairness.
- The difference between GSP-SSL and NNCLR lies primarily in their respective positive sampling strategies. However, the novelty of the proposed strategy is limited.

**Questions:**

- It would be better to make prior knowledge in an unsupervised manner, except using pretrained visual model and target dataset.
- The supervised results are supposed to be shown in Table 2.

---

> ### Author Response · Authors · 2023-11-17
> **Part 1/2**
>
> - **The employment of prior knowledge, specifically in the form of a pretrained visual model and the target dataset, diverges from the fundamental principles of Self-Supervised Learning (SSL)**
>
> *Response:* We thank the reviewer for raising that concern. While we agree with the fact that using a pretrained model as part of SSL training is a shift, we ought to emphasize that one of our core contributions is in bringing forward the importance of positive sample selection (which we often illustrated employing pretrained models). We believe that this point carries a lot of weight to guide future research direction in SSL: improving the sampling and selection of positive pairs has the potential to greatly outpace progress made around data augmentation and architecture search.
>
> Additionally, we emphasize that even when we employed, for example, CLIP pretrained embeddings for GPS, such models did not have access to labels during training. That is, the empirical result remains in the realm of SSL. We entirely agree with the reviewer’s point for the cases where we used a supervised embedding, but again, our main motivation in doing so was to demonstrate how much richer SSL representations can be with a label-informed embedding. We made sure to clarify that point and use purely SSL embeddings, i.e., CLIP, in our manuscript in the main experiments (Table 2 and 6) and removed the supervised embeddings.
>
> Furthermore, we hope to further address the reviewer’s concern with a new set of experiments (presented in Tables 8 and 9):
>
> |            |    dataset   | GPS-Backbone |  GPS-Dataset | backbone | val_acc1 |
> |------------|:------------:|:------------:|:------------:|:--------:|:--------:|
> |   VICReg   | FGVCAircraft |       -      |       -      | ResNet50 |   39.99  |
> | GPS-VICReg | FGVCAircraft |   MAE_vitL   | FGVCAircraft | ResNet50 |   42.87  |
> |   SimCLR   | FGVCAircraft |       -      |       -      | ResNet50 |   47.11  |
> | GPS-SimCLR | FGVCAircraft |   MAE_vitL   | FGVCAircraft | ResNet50 |   46.93  |
>
> Which we hope demonstrate that even when no pre-trained embedding is accessible to GPS, and a new one needs to be re-trained on the given dataset (e.g., using MAE as in the above results), its use in GPS can yield final performances at least as good as the SSL baseline (SimCLR) or even greater (VICreg).
>
> - **The incorporation of such prior knowledge raises concerns about the fairness of comparisons with existing SSL methods. There is a potential risk that the pretrained visual model and target dataset might leak additional information into the model, thereby skewing results and leading to issues of unfairness.**
>
> *Response:* This is a great observation that needs to be addressed. First (as briefly mentioned in the above point), we have conducted new experiments where none of the DNN is pretrained, which showed great benefits of the proposed strategy. Another direction that we have also explored is to see what would happen if all the methods (GPS and the SSL baselines) had access to that extra information. To that end, we have conducted an experiment where we employed SSL and GPS training on CIFAR10 and FGVCAircraft starting from a backbone that is either random (realistic setting), Imagenet pretraining, or CLIP pretrained. That is, all methods have access to the same “leaked” information. We observe that even in that setting, GPS manages to produce greater results. This added experiment also further reinforces our main point: being able to correctly select positive samples is crucial to ensure that SSL training is successful, even when starting from a near-optimal initialization.
>
> |            |              |  Cifar10 |            | FGVCAircraft |            |   |
> |------------|--------------|:--------:|:----------:|:------------:|:----------:|---|
> |            |              | Weak aug | Strong Aug | Weak aug     | Strong Aug |
> |   SimCLR   | RN50         |    46.69 |      87.39 |         5.67 |      27.36 |
> | GPS-SimCLR |              |     85.2 |      90.48 |        17.91 |      43.56 |
> |   SimCLR   | RN50_pt      |    43.99 |      94.02 |        17.91 |      59.92 |
> | GPS-SimCLR |              |     91.3 |      95.53 |        39.45 |      66.88 |
> |   SimCLR   | RN50_CLIP_pt |    45.57 |      90.26 |         6.21 |      41.04 |
> | GPS-SimCLR |              |    89.44 |      91.23 |        24.15 |      49.63 |

---

> ### Author Response · Authors · 2023-11-17
> **Part 2/2**
>
> - **The difference between GSP-SSL and NNCLR lies primarily in their respective positive sampling strategies. However, the novelty of the proposed strategy is limited.**
>
> *Response:* Our method is indeed inspired by NNCLR. However, GPS-SSL is a more general form of NNCLR. As mentioned in our general answer and in the theoretical part of our submission, GPS should be thought of as a general formulation of positive pair sampling using guidance that takes the form of prescribing the space in which nearest neighbors are sampled. That space, however, need not be given from another deep neural network but can be given from known data augmentations (thereby recovering baseline SSL), or from external knowledge such as querying an oracle. Last but not least, our formulation is identical for any SSL method employed (as opposed to NNCLR, which has been derived and tested uniquely on SimCLR). Our hope is that GPS can be seen as a general add-on akin to data augmentation that can be employed with any SSL loss.
>
> - **It would be better to make prior knowledge in an unsupervised manner, except using pretrained visual model and target dataset**
>
> *Response:* We entirely agree with that concern and have now provided numerous experiments to address it, including in the above answer to concern 1.,  and the general comment.
>
> - **The supervised results are supposed to be shown in Table 2**
>
> *Response:* We are sorry for the confusion, we have updated the caption for Tables 1 and 2 to make sure that we clearly state which method and which metric is being presented (classification accuracy).

---

> > ### Comment · Reviewer_aLRr · 2023-11-22
> > **Reviewer Response**
> >
> > Thanks for the response, which somehow addressed some of my concerns. However, the additional experiment cannot solve my concerns about the plausibility and fairness of using pretrained visual model. The novelty is also limited in comparison with NNCLR. Additionally, it's important to note that the CLIP model does have access to the labels, even though these labels might be noisy. I am afraid after taking into account these issues, I may not be able to increase my score.

---

> > > ### Author Response · Authors · 2023-11-22
> > > **New Response**
> > >
> > > Dear Reviewer aLRr,
> > >
> > > We appreciate your prompt response before the end of the discussion period, allowing us the opportunity to address your concerns. We fully understand the issue you raised regarding the fair comparison between GPS and other methods. To address this, we have conducted additional experiments. First, we applied the same amount of pretraining to all methods (GPS and SSL benchmarks), revealing superior final performances even in that setting. Second, we employed random initialization for all GPS models (both the backbone and the model for obtaining positive samples), demonstrating the competitiveness of GPS. For further details, we refer the reviewer to our previous response and the comprehensive explanation provided in the general answer regarding the results tables.
> > >
> > > In addition to resolving the fairness issue, we acknowledge the reviewer's perspective that our method may seem like an incremental improvement over NNCLR. However, we argue that our approach opens several avenues compared to NNCLR. It allows SSL to be trained on very small datasets with weak augmentations, and our formalism provably generalizes NNCLR. Specifically, we show how, in our setting, traditional SSL positive sampling can be recovered with the right embedding space—a scenario beyond the reach of NNCLR.
> > >
> > > To address these concerns, we plan to update our manuscript to explicitly state in the introduction and contributions that our method builds upon NNCLR and extends it. We will also clarify that experiments using CLIP and other pretrained backbones offer insights into GPS's capabilities in those cases. However, fair comparisons with SSL baselines can only be made when either (i) none of the models is pretrained or (ii) all models are pretrained from the same source.
> > >
> > > We hope this response aids the reviewer in evaluating our contributions, which extend beyond introducing a new SSL method to opening a new avenue of research in informed positive pair sampling. We would greatly appreciate it if they could reflect it in their score.
> > >
> > > Best regards,
> > >
> > > The authors

---

> ### Author Response · Authors · 2023-11-21
> **Reminder**
>
> Dear Reviewer aLRr,
>
> We hope that you have had a chance to consider our rebuttal. Your comments greatly helped us improve our manuscript, we sincerely thank you for that. We would like to kindly remind you that ICLR won't have a second round of discussion. As such, we would deeply appreciate it if you could raise any remaining concern or comment that you may have. This way, we will have a chance to clarify any last issue before the end of the discussion period which is now very near. If you consider your concerns addressed, then we hope that you will find time to revise your score and review accordingly.
> Thank you again for participating in the peer-review process.
>
> Best regards,
> The authors

---

### Author Response · Authors · 2023-11-17
**Part 3/3**

# Add further GPS embeddings:
To provide yet an additional comparison on using GPS with an embedding that is neither pretrained from another SSL method, nor pretrained on a different dataset, we propose a realistic setting of employing a MAE training on a ViT-large on the same dataset being considered for GPS. We observe that GPS is able to match the SSL baseline (with strong augmentations) indicating that even without any strong prior on the embedding used to sample the positive pairs, GPS is able to compete with the SSL baseline (this table in now in Table 8 and 9 on page 14 of our submission).

|            |    Dataset   | GPS-Backbone | GPS-Dataset | Backbone | Val_acc1 |
|------------|:------------:|:------------------:|:-------------:|:--------:|:--------:|
|   VICReg   | FGVCAircraft |          -         |       -       | ResNet50 |   39.99  |
| GPS-VICReg | FGVCAircraft |      MAE_vitL      |  FGVCAircraft | ResNet50 |   42.87  |
|   SimCLR   | FGVCAircraft |          -         |       -       | ResNet50 |   47.11  |
| GPS-SimCLR | FGVCAircraft |      MAE_vitL      |  FGVCAircraft | ResNet50 |   46.93  |

# Provide linear probe evaluation on the GPS embeddings:
The last comment raised by the reviewers was on providing linear probe evaluation on the embeddings used by GPS to perform the positive sampling. We have now computed them on two datasets and provide them below, that table is now in the Appendix as Table 8 and 9 on page 14. We have also added reference to it in the main text.


|                     | Cifar10	 | FGVCAircraft |
|---------------------|:-------:|:------------:|
| CLIP_RN50_LAION400M |   0.88  |     0.45     |
|  MAE_vitB_ImageNet  |   0.86  |     0.28     |
|  MAE_vitL_ImageNet  |   0.91  |     0.37     |
| vae_RN50_objects365 |   0.29  |     0.02     |

We observe that SSL and GPS SSL provide much richer representations able to outperform the GPS embedding stand-alone, with the exception of the MAE ViT-large pretrained on Imagenet and supervised RN18 and RN50 on FGVCAircraft that matches (GPS) SSL on CIFAR10.


We hope that our added experiments and updated manuscript answer all the reviewers’ concerns, and remain available for further discussion until the end of the discussion period.

---

### Author Response · Authors · 2023-11-17
**Part 2/3**

The main concerns shared by all reviewers were around the need to provide further experimental comparisons. In particular:

# Add additional backbones:
Previous version: randomly initialized Resnet18 on Cifar10, FGVCAircraft, PathMNIST, and TissueMNIST using a CLIP-pretrained RN50 for strong augmentations and ImageNet-pretrained ResNet18 for strong augmentations
New version: randomly initialized Resnet18 and Resnet50 on Cifar10, FGVCAircraft, PathMNIST, and TissueMNIST using only CLIP-pretrained RN50 for both strong and weak augmentations
GPS has (for some of our comparisons) access to additional information from the pre-trained embedding used to sample the positive pairs.

First, we ought to highlight that even when the embedding used to sample the positive pair is pre-trained, it is most of the time done in an unsupervised SSL manner. As such, GPS does not have access to any label information. That being said, we agree that unsupervised information is encoded in that embedding. However, we demonstrate in a new set of experiments that this is not the reason for the improved GPS performances–instead the performance gains come from the improved positive sampling strategy. To convey that point, we propose below to retrain the SSL baseline starting from the same backbone and consider three cases: random initialization (which we originally had), CLIP pretrained (no label information but external unsupervised data), and supervised:
|            |              |    Cifar10   |    Cifar10   | FGVCAircraft |    FGVCAircraft   |
|------------|--------------|:------------:|:----------:|:------------:|:----------:|
|            |              |   Weak aug   | Strong Aug |   Weak aug   | Strong Aug |
|   SimCLR   | RN50         |     46.69    |    87.39   |     5.67     |    27.36   |
| GPS-SimCLR |   RN50           |     85.2     |    90.48   |    17.91     |    43.56   |
|   SimCLR   | RN50_sup_pt  |     43.99    |    94.02   |    17.91     |    59.92   |
| GPS-SimCLR |    RN50_sup_pt      |     91.3     |    95.53   |    39.45     |    66.88   |
|   SimCLR   | RN50_CLIP_pt |     45.57    |    90.26   |     6.21     |    41.04   |
| GPS-SimCLR |   RN50_CLIP_pt   |     89.44    |    91.23   |    24.15     |    49.63   |



Those results are obtained using strong augmentation which we found to be the best case scenario for the SSL baseline. We observe that regardless of the additional information that both GPS and the SSL baselines share, GPS’s final performance is consistently greater. This table is now added in the main paper as Table 4 on page 9 of our submission.

# Add further comparison with greater number of epochs:
We had previously reported results where most of the models were trained for 400 epochs, and some were trained for 200 epochs. However, and as pointed out by the reviewers, many SSL methods require longer training time than 200 epochs. As such, we have implemented two changes. First, all the models in the reported tables in the submission are for 400 epochs (Tables 2 and 6). Second, we performed a few additional experiments training for 1000 epochs with a ResNet18 using SimCLR and VICReg (with and without GPS) which we report below

|            | Cifar10 |          | FGVCAircraft |          |
|------------|:-------:|:--------:|:------------:|:--------:|
|            | 400 eps | 1000 eps |    400 eps   | 1000 eps |
|   SimCLR   |  88.26  |   91.25  |     39.87    |   45.55  |
| GPS-SimCLR |  89.57  |   91.10  |     50.08    |   51.64  |
|   VICReg   |  89.34  |   90.61  |     33.21    |   41.19  |
| GPS-VICReg |  89.68  |   89.84  |     45.48    |   49.29  |

Where we employ strong augmentations (the standard SSL setting). We observe that GPS consistently improves performances and arrives on par with SimCLR on Cifar10 after 1000 epochs. We expect this result to be aligned with our observation that positive pair sampling in SSL has been greatly optimized on CIFAR and Imagenet from careful tuning of the data-augmentations, but falls short with weak data augmentation or on other datasets (as seen above). This table is now Table 3 on page 8 of our submission.

---

### Author Response · Authors · 2023-11-17
**Part 1/3**

We thank all the reviewers for their thorough review of our submission, and for taking the time to provide constructive comments. We have conducted additional experiments to answer all of the reviewers’ concerns and we strongly believe that those changes have made our submission stronger. We provide in this general answer an overview of the changes and we also have highlighted in our submission all those changes in *blue*. In addition, we provide per-reviewer answers to address specific concerns.

We would like first to remind the reviewers that our core contribution is twofold:
 - We provide a formalism to explore novel positive pair sampling strategies in Self-Supervised Learning (SSL) which recover standard SSL sampling, NNCLR (an eponymous SSL method), and extend to any user-defined sampling strategy. If anything, our hope is that exploring that direction will be considered more extensively in future avenues as our preliminary experiments strongly demonstrate potential to further push SSL performances

 - We explore novel positive pair sampling solutions guided by an external embedding space for which we explore a variety of settings. Our strategy–coined GPS for Guided Positive Sampling–demonstrates that with such improved sampling, SSL methods can learn rich representations even when the data-augmentations being employed are minimal (the reliance on data-augmentation is one of the main bottleneck of many SSL methods). We demonstrate that benefit on a variety of SSL methods (SimCLR, VICReg, BarlowTwins, BYOL) on 5 datasets and with 2 different architectures.

---

### Meta-Review · Area_Chair_FJCA · 2023-12-09

**Metareview:**

In this paper, the authors proposed a novel method for self-supervised learning. The authors use nearest neighbors in a transformed space, in which Euclidean distance corresponds to semantic relations. The reviewers see that the proposed methods have some merits, but they still find significant limitations in the paper and limited experimental results. Self-supervised learning is a well-studied topic that engines most of our current machine-learning algorithms today. The reviewers believe a stronger empirical demonstration is needed for a novel SSL procedure to be presented at a top conference. The proposed algorithm does not significantly improve baselines for strong augmentation techniques. One of the main criticisms of this paper is not using larger databases, lime imagenet, and other tasks to evaluate the quality of the proposed algorithm. The bar for novel SSL algorithms is high because the work on this topic is consequential and far-reaching.

Even though this did not come up in the reviews and has taken no role in my decision, I encourage the authors to compare their procedure with mixup. Because it is one of the leading algorithms in data augmentation, it typically provides significant improvements for data augmentation.

**Justification For Why Not Higher Score:**

The algorithm is not well-compared and the literature on SSL is so vast that better comparison are needed.

**Justification For Why Not Lower Score:**

not possible.

---

### Decision · Program_Chairs · 2024-01-16

Reject